# Tunable assembly of truncated nanocubes by evaporation-driven poor-solvent enrichment

Zhong-Peng Lv [1], Martin Kapuscinski [1] & Lennart Bergström[1]

Self-assembly of nanocrystals is extensively used to generate superlattices with long-range translational order and atomic crystallographic orientation, i.e. mesocrystals, with emergent mesoscale properties, but the predictability and tunability of the assembly methods are poorly understood. Here, we report how mesocrystals produced by poor-solvent enrichment can be tuned by solvent composition, initial nanocrystal concentration, poor-solvent enrichment rate, and excess surfactant. The crystallographic coherence and mesoscopic order within the mesocrystal were characterized using techniques in real and reciprocal spaces, and superlattice growth was followed in real time by small-angle X-ray scattering. We show that formation of highly ordered superlattices is dominated by the evaporation-driven increase of the solvent polarity and particle concentration, and facilitated by excess surfactant. Poor-solvent enrichment is a versatile nanoparticle assembly method that offers a promising production route with high predictability to modulate and maximize the size and morphology of nanocrystal metamaterials.

[1] Department of Materials and Environmental Chemistry, Arrhenius Laboratory, Stockholm University, S-106 91 Stockholm, Sweden. Correspondence and requests for materials should be addressed to L.B. (email: lennart.bergstrom@mmk.su.se)

Nanocrystals with well-defined sizes and shapes can assemble into superlattices with long-range translational order and atomic coherence that combine or transcend the intrinsic shape-and size-dependent properties and the collective properties of the interacting nanocrystals[1–7]. While spherical nanocrystals usually assemble into face-centered cubic or hexagonal-close packed arrangements, polyhedral nanocrystal assemblies display a rich structural diversity that depends on both particle shape and the anisotropic particle interactions[8–13]. Assembly methods that modulate the particle interactions, by, e.g., an evaporation-driven increase of the particle concentration[14–17], destabilization by addition of non-solvent[18–20], or by application of external fields[21,22], have been used to produce well-ordered superlattices with sizes from hundreds of nanometers to several hundred micrometers.

Assembly of anisotropic nanocrystals into large and well-ordered superlattices has been found to depend on a range of parameters, including the initial particle concentration[23,24], evaporation rate[14,18,25], and the composition and concentration of additives, in particular excess surfactant[26–28], but a systematic understanding remains to be developed. The evaporation-driven methods suffer from poor spatial control of the evaporation rate, which leads to convective flows and inhomogeneous superlattices. The destabilization methods, especially the poor-solvent diffusion technique, is limited by slow diffusion across interfaces and characterized by a low yield and poor scalability.

In this study, we demonstrate a tunable and robust evaporation-driven, poor-solvent destabilization method for reproducible and predictable assembly of oleate-capped truncated iron oxide nanocubes (NCs) into large and well-ordered superlattices with long-range translational order and atomic crystallographic orientation, also known as mesocrystals[29,30]. We investigate how the size, morphology, and degree of order of the mesocrystals depends on particle concentration, solvent polarity, growth rate, and the amount of excess oleic acid (OA). The enrichment rate of the poor solvent (PS) is tuned by membrane-controlled evaporation/diffusion of the good solvent (GS). Time resolved small-angle X-ray scattering (SAXS) in levitating drops provide insight into the structural evolution during self-assembly and confirm that nanocrystal assembly by poor-solvent enrichment proceeds similarly in bulk liquids and small droplets. We show that the onset and duration of self-assembly could be predicted from the evaporation-induced increase of the solvent polarity and nanocrystal concentration. The tunable evaporation-driven poor-solvent enrichment (EDPSE) assembly method enables predictive production of large and well-ordered nanocrystal superlattices for optoelectronic, magnetic, and biomedical applications.

## Results

**Overview of the EDPSE assembly process**. We have used a poor-solvent enrichment method to grow very large mesocrystals by controlled assembly of oleate-capped truncated iron oxide NCs with a narrow size distribution with a standard deviation of 5.5% and average edge length of $d_{TEM} = 10.8$ nm (Fig. 1a and Supplementary Fig. 1). The assembly studies were performed on highly purified NCs that had been repeatedly washed in hexane or toluene and 1-pentanol (see Methods, Supplementary Note 1 and Supplementary Fig. 2). The NCs were dispersed in solvent mixtures of a good, low polarity solvent (GS) with a relatively high vapor pressure, and a poor, high-polarity solvent (PS) with a relatively low vapor pressure. Self-assembly studies have been performed on bulk dispersions in a vessel and on confined levitating dispersion drops. The relative evaporation rate and thus the time-dependent increase of the polarity of the solvent mixture was controlled by the difference in vapor pressure of the GS and

PS and the diffusion rate through one or several layers of a polyethylene (PE) membrane that covered the opening of the vessels (Fig. 1a).

EDPSE assembly can be used to repeatedly produce large and well-ordered superlattices. The SEM image, corresponding high-resolution SEM (HRSEM) image and Fast-Fourier Transformation (FFT) pattern of the superlattice in Fig. 1b shows that EDPSE assembly of truncated NCs can yield superlattices with a size of several hundred microns with long-range translational order and atomic crystallographic orientation, i.e., mesocrystals[30]. The EDPSE method is also able to produce well-ordered superlattices in confined space, e.g., in a levitating drop, as shown by the well-defined one-dimensional (1D) SAXS profile that suggests that the NCs assembled in an *fcc* lattice (Fig. 1c).

Indeed, some of the mesocrystals produced by EDPSE assembly are sufficiently large allowing them to be investigated by single-crystal X-ray diffraction (SC-XRD) (Fig. 1d). The two-dimensional (2D) SC-XRD pattern of the single mesocrystal in Fig. 1e exhibits well-defined diffraction arcs. Indexing the SC-XRD pattern confirmed the inverse spinel atomic structure of the NCs (Fig. 1e). The single crystal-like diffraction pattern as the mesocrystal is rotated (Supplementary Movie 1) indicates a long-range order on the atomic-scale in all three dimensions of the assembled NCs[30]. The presence of diffraction arcs and not diffraction spots suggests that the NCs are slightly misaligned[31,32], which probably is caused by the NCs being slightly tilted with respect to the main crystallographic direction.

Assembly of large and well-ordered mesocrystals by the EDPSE method requires optimization of the solvent composition and initial NC concentration and that the enrichment rate of the poor-solvent is controlled. Hexane and 2-propanol were selected from a range of GSs and PSs as the optimal GS and PS pair, mainly due to their mutual miscibility, and large difference in vapor pressure and polarity. The low toxicity and relatively low costs of these solvents were also important features of relevance for large-scale applications. Unconstrained evaporation of pure hexane is about 10 times faster than pure 2-propanol, with the evaporation rate decreasing with time due to the cooling caused by the rapid, unconstrained evaporation (Supplementary Fig. 3a). Covering the vessel with one or several layers of the PE membrane not only reduced the evaporation rate but also increased the difference in evaporation rate of hexane and 2-proponal to 200 because hexane diffuses much faster than 2-propanol through the hydrophobic PE membrane (Supplementary Fig. 3b–d). The evaporation rates of pure solvents in vessels covered with one or several PE layers did not vary over time, which suggests that the evaporation is sufficiently slow to be considered isothermal. The removal of solvent from the dispersions is thus driven by the evaporation but the rate is primarily controlled by the diffusion rate of hexane through the PE membrane. It should be noted that covering the vessel with one or several PE membranes provides a significantly larger enhancement and better tunability of the differential evaporation rate of the two solvents compared to, e.g., gas bubbling[33] or reduction of the pressure[18,20,25,34].

The mesocrystal growth process can be evaluated from the position of the black mesocrystal ring on the vessel wall, as shown in Fig. 2a. The initial liquid level was $H(0)$ where the total liquid volume $V(0) = V_{GS}(0) + V_{PS}(0) = 15$ mL. The mesocrystals started to form when the liquid level reached $H(t_1)$, and the major mesocrystal growth process ends when the liquid level reached $H(t_2)$ (Fig. 2a). The differential evaporation rate between the good and PS is about 200, hence, the evaporation rate of the GS ($v_{GS}$) in the mixed solvent could

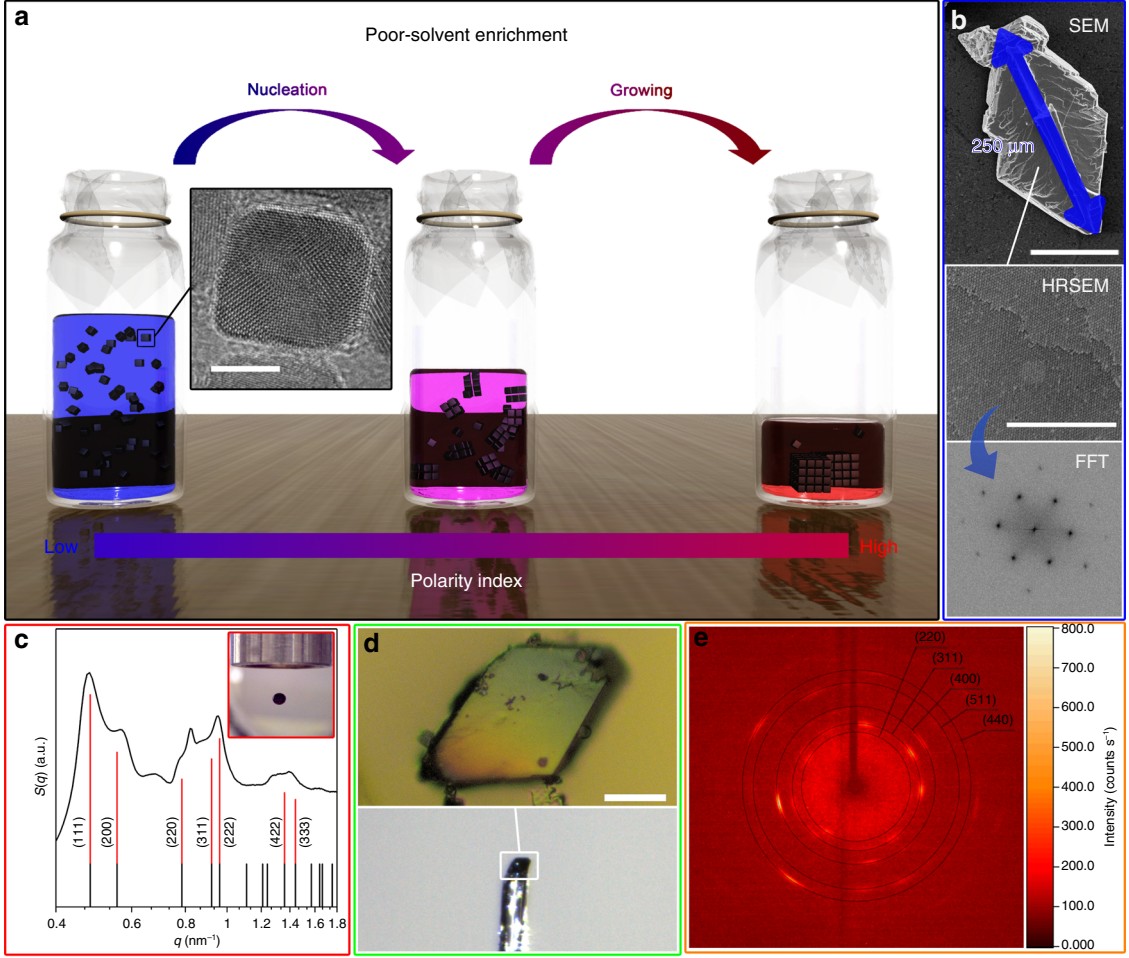

**Fig. 1** Large and well-ordered mesocrystals assembled by evaporation-driven poor-solvent enrichment. **a** Schematic illustration of the setup for assembly of nanocrystals into mesocrystals by evaporation-driven poor-solvent enrichment in a vessel. Inset: HRTEM image of a truncated nanocube. Scale bar: 5 nm. **b** SEM image of a large and well-ordered rhombic mesocrystal prepared by poor-solvent enrichment (sample #9, Table 1); with HRSEM image and corresponding FFT pattern of the (110) facet. Scale bar = 100 µm for SEM and =500 nm for HRSEM **c** The structure factor $S(q)$ (black curve), obtained from a 1D SAXS curve, of mesocrystals generated by evaporation-driven poor-solvent enrichment in a levitating drop (inset) (sample #D1, Table 2) with indexed reflections of an *fcc* superlattice (red long vertical lines and black short vertical lines indicate the visible and allowed indices of an *fcc* superlattice, respectively). **d** Optical image of a single mesocrystal (sample #9, Table 1) (top) mounted on a Bruker diffraction meter (bottom). Scale bar = 20 µm (top). **e** 2D SC-XRD pattern of the single mesocrystal in (**d**), with the position of the strongest diffraction indices of magnetite marked by orange auxiliary circles. The intensity is given by the color scale bar

be estimated using,

$$v_{GS} = \frac{dV_{GS}}{dt} = v^*_{GS} \frac{V_{GS}(t)}{V_{GS}(t) + 0.283 \times V_{PS}(0)} \quad (1)$$

where $v^*_{GS}$ is the evaporation rate of pure GS, $V_{GS}(t)$ is the time-dependent volume of GS and $V_{PS}(0)$ is the initial volume of 2-propanol. Detailed information on the derivation of the mathematical expressions and evaluation of the analytical solution of $V_{GS}(t)$ with different initial GS to PS volume ratio ($V_{GS}/V_{PS}$) and/or number of PE layers ($N_{PE}$) values are given in the Supplementary Note 2 and Supplementary Table 1. The critical NC concentration $c_C = c(t_1)$ when mesocrystal formation is initiated is directly related to $H(t_1)$. The time of the major growth process $\Delta t = t_2 - t_1$ could be estimated from the width of the ring (i.e., the difference of $H(t_2)$ and $H(t_1)$) and Eq. 1 (see also Supplementary Note 3). Using the above model, systematic studies of the mesocrystal growth process were performed at different GS to PS volume ratios, $V_{GS}/V_{PS}$; initial particle concentration, $c(0)$; number of PE layers $N_{PE}$; and ratio

of the volume of added OA to NCs powder weight, $V_{OA}/m_{NC}$ (Table 1).

**Effect of GS to PS volume ratio ($V_{GS}/V_{PS}$) on EDPSE assembly.** Figure 2a shows that an increase of $V_{GS}/V_{PS}$ from 2.5 to 12.5 results in a reduction in $\Delta t$ from 19 to 5 h, and a corresponding increase of the critical particle concentration for the onset of mesocrystal growth from 3.9 to 13 mg mL$^{-1}$ (samples #1–3, Table 1). The time-dependent polarity $P(t)$ can be estimated from the change in composition:

$$P(t) = \frac{(P_{GS} \times V_{GS}(t) + P_{PS} \times V_{PS}(0))}{V_{GS}(t) + V_{PS}(0)} \quad (2)$$

where the polarity index $P_{GS}$ and $P_{PS}$, defined from the widely used Snyder's polarity index, are 0.100 and 3.90 for hexane and 2-propanol, respectively[35,36]. The critical polarity $P_C = P(t_1)$, for the onset of mesocrystal growth, decreased from 1.51 to 1.31 with increasing $V_{GS}/V_{PS}$. It is interesting to note that mesocrystal growth in a dispersion in only hexane (sample #4, Table 1), where

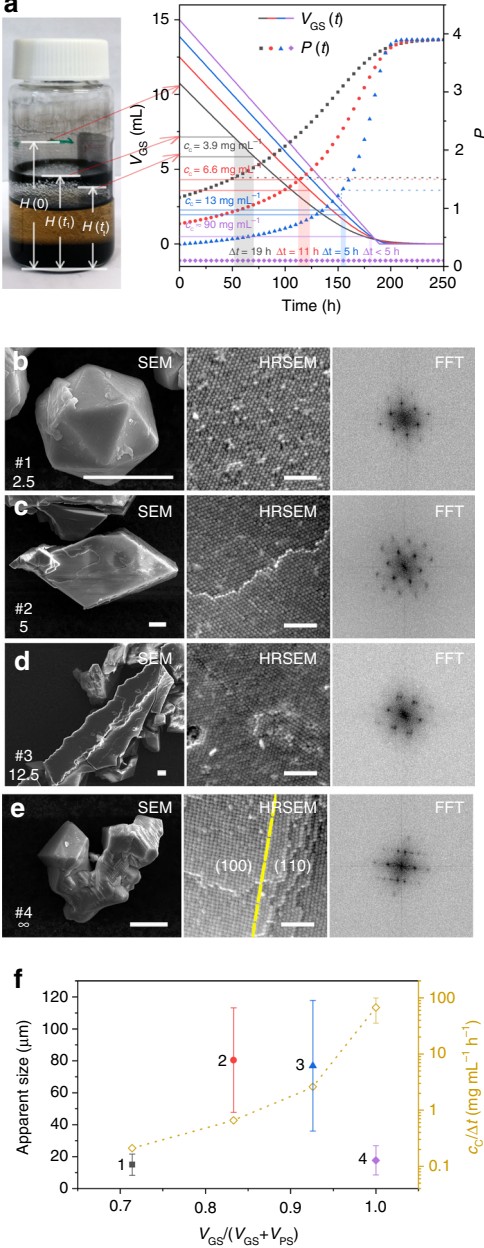

**Fig. 2** Assembly of mesocrystals grown at different $V_{GS}/V_{PS}$ ratios by evaporation-driven poor solvent enrichment. **a** Left: optical image of the vessel after mesocrystal formation is complete (sample **#1**); the initial liquid level and the height for the onset and completion of mesocrystal growth are indicated as $H(0)$, $H(t_1)$, and $H(t_2)$, respectively. Right: the $V_{GS}(t)$ (solid lines) and $P(t)$ (dotted lines) curves for $V_{GS}/V_{PS} = 2.5$, $N_{PE} = 1$ (black square), $V_{GS}/V_{PS} = 5$, $N_{PE} = 1$ (red filled circle), $V_{GS}/V_{PS} = 12.5$, $N_{PE} = 1$ (blue triangle), and $V_{GS}/V_{PS} = \infty$, $N_{PE} = 1$ (purple diamond). The auxiliary lines and shaded regions show $\Delta t$ and $c_C$, estimated from the position of $H(t_1)$ and $H(t_2)$. Scanning electron microscopy (SEM) images for individual mesocrystals, high-resolution SEM (HRSEM) of the mesocrystal facets and corresponding fast-Fourier transform (FFT) patterns of mesocrystals grown from dispersions at $V_{GS}/V_{PS}$ ratios of: **b** 2.5; **c** 5; **d** 12.5; **e** ∞. Scale bar = 10 μm for all SEM images and = 100 nm for all HRSEM images. **f** Apparent size of mesocrystals (filled symbols) and $\frac{c_C}{\Delta t}$ (open diamonds) as a function of $V_{GS}/(V_{GS} + V_{PS})$. Error bars for the size represent the standard deviation of the mean (number of measurements: 290 for #1, 145 for #2, 118 for #3, and 306 for #4)

the $P$ value is always 0.1, resulted in rapid mesocrystal growth ($\Delta t$ significantly less than 5 h) and a $c_C$ around 90 mg mL$^{-1}$.

The mesocrystals exhibited a rich range of morphologies including quasi-icosahedron, five-armed star, octahedron, triangle, and rhombus (Fig. 2b–e). The twinned structures (quasi-icosahedron and five-armed star) were primarily observed in the small mesocrystals produced at $V_{GS}/V_{PS}$ ratio of 2.5 (Fig. 2b) while the rhombic and triangular single domain structures dominated in mesocrystals produced at a $V_{GS}/V_{PS}$ ratio of 5 (Fig. 2c). The HRSEM images and FFT patterns of the icosahedral and rhombic mesocrystals show a highly ordered NC alignment with a $c2mm$ symmetry on the (110) facet (Fig. 2b,c). The mesocrystals produced at $V_{GS}/V_{PS}$ ratios of 12.5 (Fig. 2d) and 5 (Fig. 2c) have a similar size but the mesocrystals in Fig. 2d display a more irregular needle-like structure with defects and distortions clearly visible on its facet and the corresponding FFT patterns. The mesocrystals produced at $V_{GS}/V_{PS}$ ratios of infinity (Fig. 2e) exhibit complex twinned structures with different orientations. The boundary between the coplanar (100) and (110) facets that is marked by yellow dash line in the HRSEM image (Fig. 2e) may be a consequence of the rapid nucleation and growth in this system. Additional structural information is given in Supplementary Fig. 4. Low magnification SEM was used to determine the apparent size (Fig. 2f), and size distribution (Supplementary Fig. 5), of mesocrystals formed from dispersions of different $V_{GS}/V_{PS}$ ratios. Figure 2f and Fig. 2c,d show that large and high-quality mesocrystals can be grown by EDPSE at $V_{GS}/V_{PS}$ ratio around 5–12.5 when $V_{OA}/m_{NC}$ is 1.6 μL mg$^{-1}$, and $c(0)$ is 3 mg mL$^{-1}$. Figure 2f also shows that the ratio of the critical concentration and time for mesocrystal growth, $\frac{c_C}{\Delta t}$, increases continuously with the $V_{GS}/V_{PS}$ ratio and the optimum range for mesocrystal growth at $V_{OA}/m_{NC} = 1.6$ μL mg$^{-1}$, and $c(0) = 3$ mg mL$^{-1}$ is around 10 mg mL$^{-1}$ h$^{-1}$. The systematic study also suggest that the major reason limiting the growth of large, high-quality mesocrystals by simple (good-solvent) drop casting[25,37–40] is the (too) rapid growth and very high critical concentration.

**Effect of initial NC concentration ($c(0)$) and number of PE layers ($N_{PE}$) on EDPSE assembly.** We have also investigated how the size and morphology of the mesocrystals depend on the initial NC concentration, $c(0)$, and differential evaporation rate. Figure 3a shows that the mesocrystal apparent size increases from 9 to 80 μm and $\frac{c_C}{\Delta t}$ increases from 0.25 to 0.6 mg mL$^{-1}$ h$^{-1}$ with increasing $c(0)$ (see Supplementary Fig. 6 for more details). Increasing $c(0)$ from 0.375 to 3 mg mL$^{-1}$, results in an increase of the $c_C$ from 1.0 to 6.6 mg mL$^{-1}$ and $\Delta t$ from 4 to 10 h, respectively (see Supplementary Table 2).

The relatively small mesocrystals formed from dispersions at $c(0) = 0.375$ mg mL$^{-1}$ display a well-defined morphology, e.g., five-armed star, icosahedron, and octahedron (Fig. 3b and Supplementary Fig. 7a). The smooth facets and sharp edges of these mesocrystals suggests that mesocrystal growth at low particle concentration proceeded with insignificant influence of other mesocrystals or the walls of the container[23]. Mesocrystals grown from dispersions at $c(0) = 3$ mg mL$^{-1}$ were significantly larger than the mesocrystals grown at a lower $c(0)$ (Fig. 3b and Supplementary Fig. 7b–d). The facets became more uneven and the icosahedron that dominated at low $c(0)$ transformed into quasi-icosahedrons without vertices. The structural transition is probably due to strain release of the twinned structure (Supplementary Fig. 8)[41]. Mesocrystals with regular twinned structures such as five-armed star and icosahedron were easier to find in samples grown at lower $\frac{c_C}{\Delta t}$, irrespective if the mesocrystals were produced from dispersions with different $c(0)$ (Supplementary Fig. 7) or $V_{GS}/V_{PS}$ ratios (Supplementary Fig. 4a, b). Hence,

**Table 1 Assembly of mesocrystals by EDPSE at different initial growth parameters**

| Sample # | $V_{GS}/V_{PS}$ | $c(0)$ (mg mL$^{-1}$) | $N_{PE}$ | $V_{OA}/m_{NC}$ (µL mg$^{-1}$) | Additional comments[a] |
|---|---|---|---|---|---|
| 1 | 2.5 | 3.0 | 1 | 1.6 | $\frac{V_{GS}}{V_{GS}+V_{PS}} \approx 0.714$ |
| 2 | 5 | 3.0 | 1 | 1.6 | $\frac{V_{GS}}{V_{GS}+V_{PS}} \approx 0.833$ |
| 3 | 12.5 | 3.0 | 1 | 1.6 | $\frac{V_{GS}}{V_{GS}+V_{PS}} \approx 0.926$ |
| 4 | ∞ | 3.0 | 1 | 1.6 | $\frac{V_{GS}}{V_{GS}+V_{PS}} = 1$ |
| 5 | 5 | 0.375 | 1 | 1.6 | |
| 6 | 5 | 0.75 | 1 | 1.6 | |
| 7 | 5 | 1.5 | 1 | 1.6 | |
| 8 | 5 | 3.0 | 2 | 1.6 | |
| 9 | 5 | 3.0 | 4 | 1.6 | |
| 10 | 5 | 3.0 | 4 | 1.6 | Scale down: total volume 1.5 mL, $A_O = 0.62$ cm$^2$ |
| 11 | 5 | 3.0 | 1 | 1.6 | Scale up: total volume 120 mL, $A_O = 7$ cm$^2$ |
| 12 | 2.4 | 5.3 | 4 | 1.6 | GS = 6 mL and PS = 2.5 mL |
| 13 | 5 | 3.0 | 1 | 0 | |
| 14 | 5 | 3.0 | 1 | 0.2 | |
| 15 | 5 | 3.0 | 1 | 0.4 | |
| 16 | 5 | 3.0 | 1 | 0.8 | |
| 17 | 5 | 3.0 | 1 | 3.2 | |
| X | 4 | 8.0 | 1 | 1.6 | |

[a]GS and PS for all samples were hexane and 2-propanol respectively without special mention. The total volumes is 15 mL and the opening area of the vessel $A_O$ is 2 cm$^2$ except for samples #10–12

optimization of $\frac{c_C}{\Delta t}$ by control of, e.g., $V_{GS}/V_{PS}$ or $c(0)$ could inhibit or minimize the formation of twinned mesocrystals.

Increasing the number of PE layers, $N_{PE} = 1$, 2, and 4 (corresponding to samples #2, #8, and #9, respectively), resulted in an increase of the duration of mesocrystal growth; $\Delta t$ increased from 11 h to 22 h with increasing $N_{PE}$ while $c_C$ and $P_C$ were unaffected (see Supplementary Table 2). Figure 3c shows that the apparent size of the mesocrystals increased from 80 to 110 µm as the $\frac{c_C}{\Delta t}$ decreased from 0.6 to 0.3 mg mL$^{-1}$ h$^{-1}$ with increasing $\Delta t$ (see Supplementary Fig. 9 for size distribution). The SEM images at different magnification in Fig. 3d show a very large mesocrystal with an apparent size of about 100 µm with well-defined facets without boundaries or distortions on the (110) facet that was grown at a very low growth rate (sample #9, $N_{PE} = 4$). The mesocrystals grown using $N_{PE} = 1$, 2, and 4 all displayed a $c2mm$ symmetry, as shown by the FFT patterns at different magnifications in Fig. 3e–h. This shows that the slow and tunable growth by EDPSE in containers covered with PE membranes is essential for the assembly of single domain mesocrystals with a coherent structure over a length scale of several tens of microns. Additional morphological information is given in Supplementary Fig. 10.

Assembly of nanoparticle superlattices by PS destabilization, using, e.g., liquid/gas phase diffusion with a two or three-layer set-up, is slow (often taking several weeks), difficult to scale up and modulate[23,31,41–44]. In contrast, the EDPSE method presented in this study is scalable (sample #10–11, Table 1) and it is also possible to reduce the process time (sample #12, Table 1) by optimization of the initial solvent composition using the evaporative model, Eq. 1 (for details see Supplementary Note 4, Supplementary Figs. 11–13).

**Effect of added OA on EDPSE assembly.** Previous reports have suggested that the addition of amphiphilic molecules such as OA can have an important effect on the assembly process and it has been speculated that the effect can be related to an improved colloidal stability[26,45,46], reduced evaporation rate[28,43], and an OA-induced interparticle depletion attraction[37,47]. We have performed a study to clarify how the amount of additional OA influences the EDPSE assembly process. We have investigated how the size and order of mesocrystals depends on the ratio of the volume of added OA to NCs powder weight, $V_{OA}/m_{NC}$

(Table 1). Figure 4a shows that the apparent size of the mesocrystals increased from 20 µm at $V_{OA}/m_{NC} = 0$ µL mg$^{-1}$ to ~80 µm at $V_{OA}/m_{NC} \geq 1.6$ µL mg$^{-1}$ (the size distribution is given in Supplementary Fig. 14a).

Measurements of the electrophoretic mobility of dilute dispersions of oleate-capped iron oxide NCs (Fig. 4b) show that the zeta potential/electrophoretic mobility ($\mu_e$) decreased with increasing amount of added OA; from 43 mV/$0.33 \times 10^{-4}$ cm$^2$ V$^{-1}$ s$^{-1}$ at $V_{OA}/m_{NC} = 0$ µL mg$^{-1}$ to about 15 mV/$0.11 \times 10^{-4}$ cm$^2$ V$^{-1}$ s$^{-1}$ at $V_{OA}/m_{NC} \geq 0.2$ µL mg$^{-1}$ (see Supplementary Fig. 14b). The electrical charge ($Z$, in units of $e$) of the NC in a low dielectric solvent can be estimated from $\mu_e$ for an equivalent sphere as [48]:

$$\mu_e = \frac{Ze}{3\pi\eta d_H} \quad (3)$$

where $\eta$ is the viscosity of the solvent. With $\eta = 0.536$ mPa s and $d_H = 20$ nm, we yield $\mu_e \approx 0.15 \times 10^{-4}$ $Z$ cm$^2$ V$^{-1}$ s$^{-1}$. This $\mu_e$ value suggests that the charge on the dispersed, oleate-capped NCs decreased from $2e$ at $V_{OA}/m_{NC} = 0$ µL mg$^{-1}$ to $1e$ at $V_{OA}/m_{NC} \geq 0.2$ µL mg$^{-1}$ and the charge distribution also became more narrow at higher OA additions (Fig. 4b). The electrophoretic measurements thus suggest that the electrostatic repulsion decreases with increasing OA addition. TG measurements of the purified NC powder (Supplementary Fig. 2b) suggests that the OA coverage on the NP surface is 2.0 molecules nm$^{-2}$, which agrees well with previous reports[49]. Previous work has shown that the ligand coverage of oleate-capped nanoparticles depends on the OA concentration in solution, and can increase to 3.5 molecules nm$^{-2}$ at OA concentration in solutions of 15 mmol or higher[50,51]. The resulting decrease of the free OA concentration in the dispersion is for most of the investigated dispersions very small. For $V_{OA}/m_{NC} = 1.6$ µL mg$^{-1}$ (the ratio of the volume of added OA to NCs powder weight in most of the investigated dispersions, see Tables 1 and 2), the estimated loss of the excess ligand concentration is only 5% or less. In the dispersions where a relatively small amount of OA was added (samples #14–16), we estimate that adsorption reduces the amount of free OA with 25–10%, respectively. Dynamic light scattering (DLS) shows that the NC hydrodynamic diameter $d_H$ is unaffected by OA addition (Supplementary Fig. 14c), which suggest that the stability is sufficient to avoid uncontrolled aggregation.

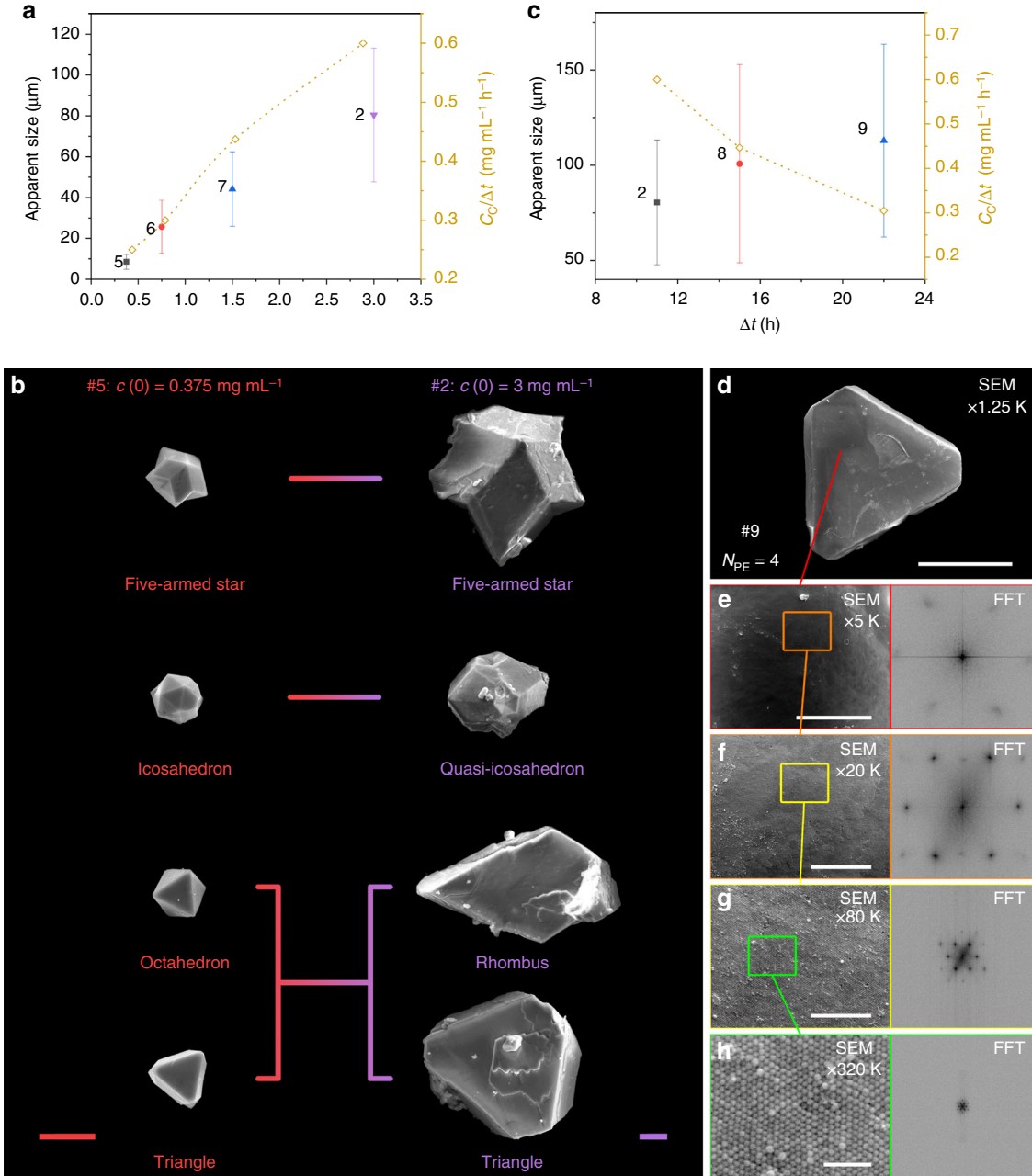

**Fig. 3** Assembly of mesocrystals by evaporation-driven poor-solvent enrichment at different NC concentrations and evaporation rate. **a** Apparent size of mesocrystals (filled symbols) and $c_C/\Delta t$ (open diamonds) as a function of initial NC concentration, $c(0)$. The error bars for the size represent the standard deviation of the mean (number of measurements: 397 for #5, 241 for #6, 120 for #7, and 145 for #2). **b** SEM images of mesocrystals produced at different $c(0)$: 0.375 mg mL$^{-1}$ (sample #5, red) and 3 mg mL$^{-1}$ (sample #2, purple). Scale bar = 10 μm for both samples. **c** Apparent size of mesocrystals (filled symbols) and $c_C/\Delta t$ (open diamonds) as a function of $\Delta t$. The error bars for the size represent the standard deviation of the mean mean (number of measurements: 145 for #2, 106 for #8, and 92 for #9). Individual mesocrystal produced at very low evaporation rate (sample #9); **d** SEM image at 1.25 k magnification, scale bar = 50 μm; and sequentially magnified (×4, 16, 64, and 256 times) SEM images with corresponding FFT patterns; **e** ×4 (5 k magnification), scale bar = 10 μm; **f** ×16 (20 k magnification), scale bar = 2 μm; **g** ×64 (80 k magnification), scale bar = 500 nm; and **h** ×256 (320 k magnification), scale bar = 100 nm

Figure 4c–h shows SEM, HRSEM, and FFT images of mesocrystals produced from dispersions at different $V_{OA}/m_{NC}$ values. The mesocrystals produced at $V_{OA}/m_{NC} = 0$ μL mg$^{-1}$ (Fig. 4c and Supplementary Fig. 15a) were relatively small and irregular with an apparent size of 20 μm while mesocrystals produced at $V_{OA}/m_{NC} \geq 0.8$ μL mg$^{-1}$ exhibit well-defined rhombus or quasi-rhombus shapes with 60–100 μm apparent sizes (Fig. 4f–h and Supplementary Fig. 15d–f).

The HRSEM images and corresponding FFT patterns in Fig. 4c–h clearly show that the crystallographic register and NC alignment improves with increasing $V_{OA}/m_{NC}$ values, from essentially random in sample #13 ($V_{OA}/m_{NC} = 0$ μL mg$^{-1}$) to highly ordered in samples #2 and #17 ($V_{OA}/m_{NC} \geq 1.6$ μL mg$^{-1}$). Hence, we find that the reduction of NC surface charge with increasing OA addition is related to the formation of larger mesocrystals with clearly defined crystal habits (Fig. 4c–h). We

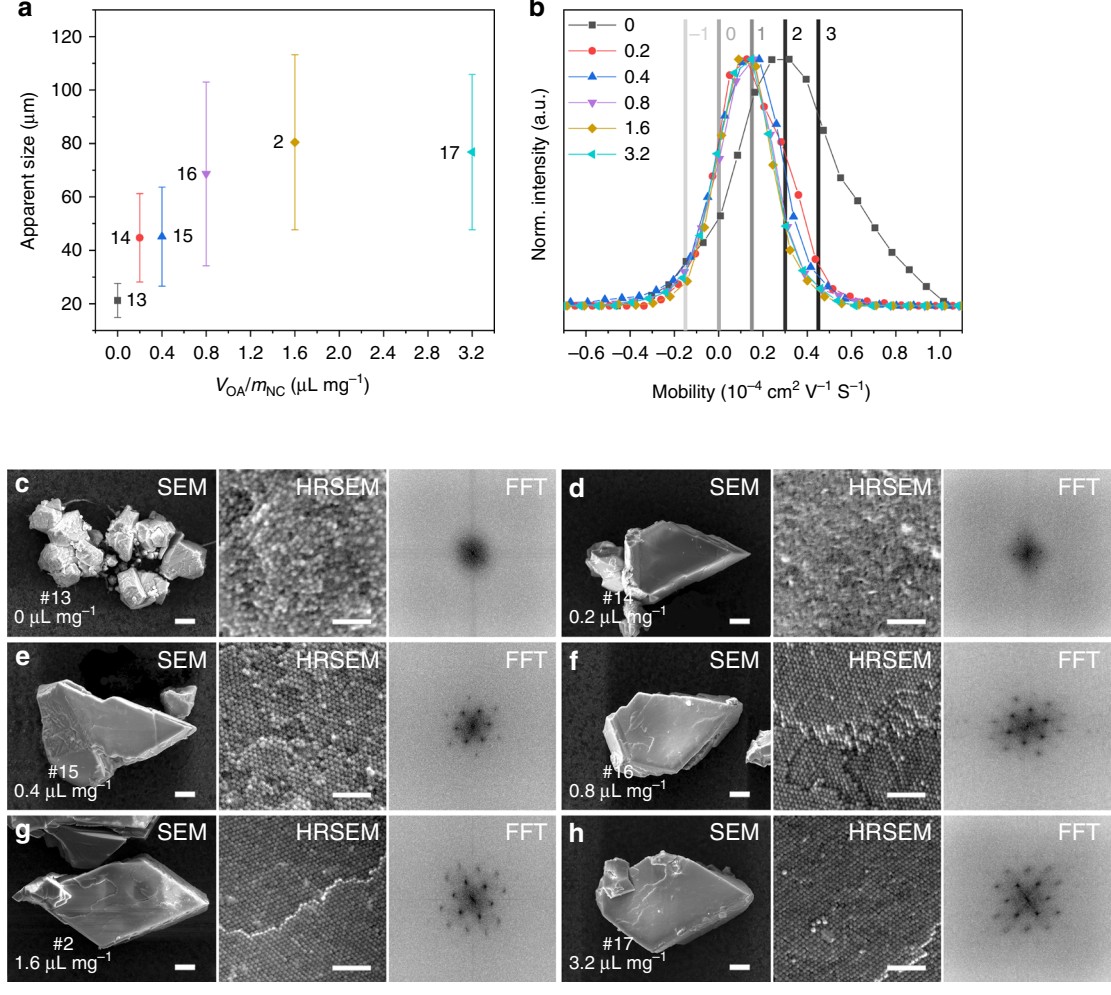

**Fig. 4** Assembly of mesocrystals by evaporation-driven poor-solvent enrichment at different amounts of added oleic acid. **a** Apparent size of mesocrystals grown from dispersions at different $V_{OA}/m_{NC}$. Error bars represent the standard deviation of the mean (number of measurements: 179 for #13, 215 for #14, 177 for #e, 113 for #16, 145 for #2 and 78 for #17). **b** Electrophoretic mobility of oleate-capped iron oxide nanocubes in chloroform at different $V_{OA}/m_{NC}$ (color lines with symbols, in unit of μL mg⁻¹). The vertical lines indicate the electrical charge (in unit of e). **c–h** Scanning electron microscopy (SEM) images for individual mesocrystals, high-resolution SEM (HRSEM) of the mesocrystal facets and corresponding fast-Fourier transform (FFT) patterns of mesocrystals grown from dispersions at $V_{OA}/m_{NC}$ of: **c** 0 μL mg⁻¹ (sample #13); **d** 0.2 μL mg⁻¹ (sample #14); **e** 0.4 μL mg⁻¹ (sample #15); **f** 0.8 μL mg⁻¹ (sample #16); **g** 1.6 μL mg⁻¹ (sample #2); **h** 3.2 μL mg⁻¹ (sample #17). Scale bar = 10 μm for all SEM images and =100 nm for all HRSEM images

**Table 2 Assembly of mesocrystals by EDPSE in a levitating droplet at different $V_{GS}/V_{PS}$ and $V_{OA}/m_{NC}$**

| Sample # | $V_{GS}/V_{PS}$[a] | $V_{OA}/m_{NC}$ (μL mg⁻¹) | $c(0)$ (mg mL⁻¹) | Equivalent bulk sample # (Table 1) |
|---|---|---|---|---|
| D1 | 5 | 1.6 | 3 | 2 |
| D2 | ∞ | 1.6 | 3 | 4 |
| D3 | 5 | 0 | 3 | 13 |

[a]GS and PS for all the samples were octane and 1-pentanol, respectively

speculate that the smaller size and poorer order of mesocrystals produced from dispersions with low amount of added OA is related to the larger surface charge distribution compared to dispersions with high amounts of added OA. It is possible that the NCs with low surface charges assemble first, while NCs with high surface charges remain in the dispersion and assemble only when the particle concentration has become very high, which may result in the formation of a less ordered surface layer. It should be noted that depletion attraction is expected to be insignificant at the relatively low amounts of free OA in the solution (the free OA volume fraction $x_{OA}$ at the onset of mesocrystal formation is always less than 2.1%, see Supplementary Table 2). However, the $x_{OA}$ in a dispersion that only contain GS (sample #4) was as high as 15.1% (see Supplementary Table 2), which suggest that the assembly in this (non-EDPSE) system could be driven by attractive depletion forces caused by the high OA concentration in solution rather than by the increasing volume fraction of the NCs[37].

**Real-time SAXS study of EDPSE assembly on a levitating droplet.** The evaporation-driven mesocrystal growth process has been followed by time-resolved SAXS measurements of a shrinking levitating colloidal droplet (Fig. 5a). The time-resolved study on the levitating droplet was performed on a solvent pair with a slow evaporation rate so that the dynamic assembly process could be followed in detail; we used octane/1-pentanol instead of hexane/2-propanol (Table 2). The time-dependent NC concentration within the droplet, $c_{DL}(t)$, the partial scattering

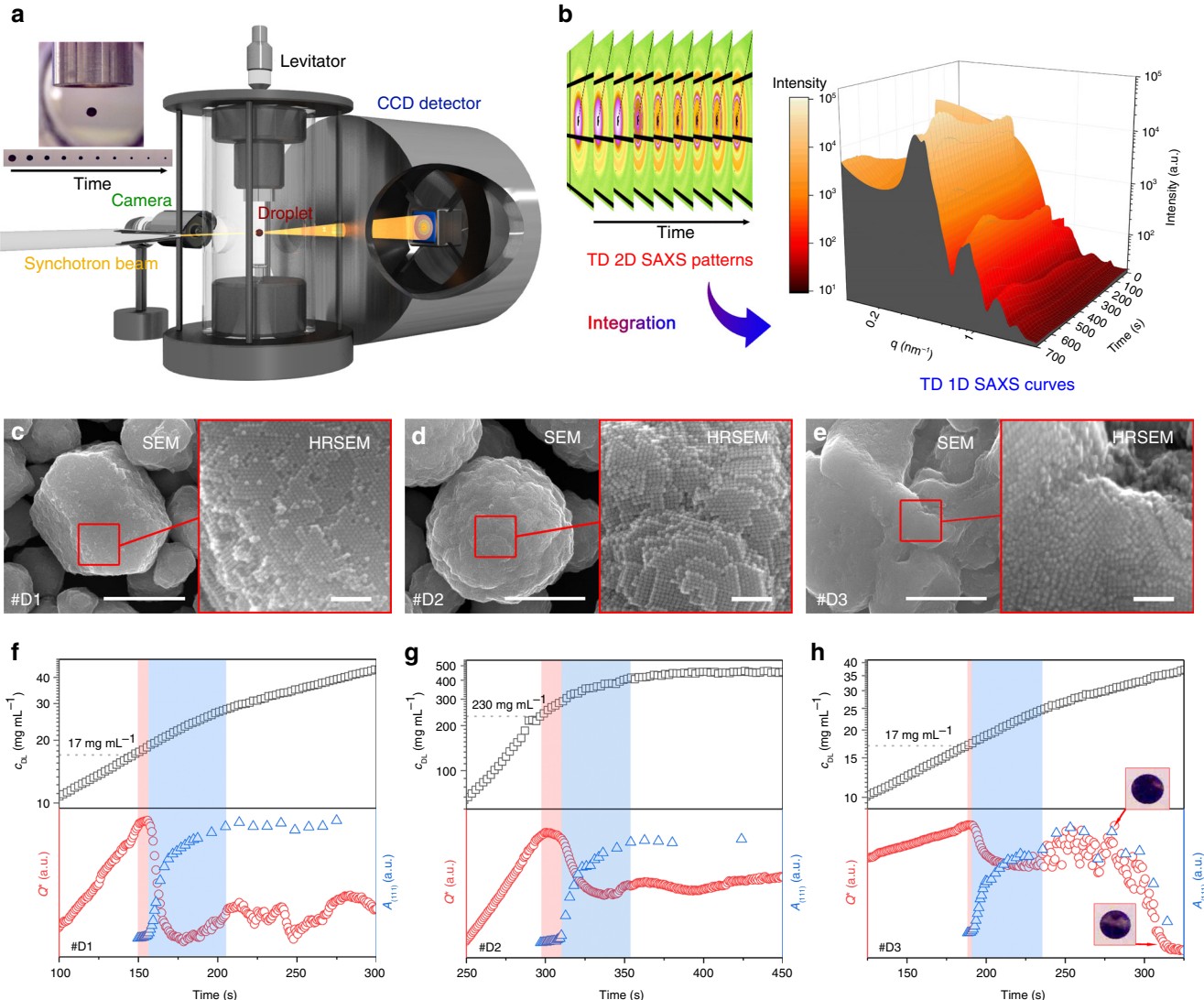

**Fig. 5** Time-resolved SAXS measurements of assembly of mesocrystals by evaporation-driven poor-solvent enrichment in a levitating droplet. Schematic illustrations of: **a** the real-time SAXS measurement of a levitating droplet and; **b** the processing of the scattering data. **c–e** Scanning electron microscopy (SEM) and high-resolution SEM (HRSEM) images of the mesocrystals formed in the levitating drops of: **c** sample #D1; **d** sample #D2; **e** sample #D3. Scale bar = 1 μm for all SEM images and = 100 nm for all HRSEM images. **f–h** Time-resolved scattering data, $c_{DL}(t)$ (black open squares), $Q^*(t)$ (red open spheres), and $A_{(111)}(t)$ (blue open triangles) for: **f** sample #D1; **g** sample #D2; **h** sample #D3. The insets in (**h**) show droplet images at $t = \sim 270$ s and $\sim 310$ s. The entire data set is displayed in Supplementary Fig. 18. The red and blue marked areas indicate the periods of nucleation/early crystal growth and the major crystal growth stage, respectively

invariant $Q^*(t)$, the (111) peak area $A_{(111)}(t)$, and the corresponding separation distance, $d_{(111)}(t)$, were extracted from the video of the shrinking droplet and the scattering curves, respectively (Fig. 5a, b, see "Methods").

Assembly of oleate-capped NCs by EDPSE in levitating droplets with a GS/PS ratio of 5 and $V_{OA}/m_{NC} = 1.6$ μL mg$^{-1}$ (sample #D1, Table 2) resulted in mesocrystals that display single domains with a feature size around 1 μm (Fig. 5c) characterized by sharp scattering peaks (Supplementary Fig. 16a), while assembly from droplets containing only GS (#D2) exhibited spherical shapes with small twinned domains (Fig. 5d), characterized by wide and overlapping peaks (Supplementary Fig. 16b). The mesocrystals assembled in droplets without added OA (#D3) display disordered surfaces (Fig. 5e) but sharp scattering peaks (Supplementary Fig. 16c). Fig. 5c–e shows that although the mesocrystals formed by EDPSE in the shrinking droplets are smaller and contain more defects compared to the

mesocrystals formed in the PE-covered vessel, it is clear that assembly in the levitating drops generates mesocrystals with similar morphologies and NC alignments as observed in corresponding samples #2, #4, and #13 (Table 1). The 1D SAXS curves of the dry beads indicated an *fcc*-structure where the NCs are tilted by 45° in the *x*, *y*, and *z* directions, and partly overlapping in a face-to-face arrangement along the [111] superlattice direction (see Supplementary Fig. 17 and Supplementary Note 5)[38]. The crystalline quality of the mesocrystals produced from dispersions containing PS (samples #D1 and #D3, Table 2) was significantly higher than the mesocrystals produced from the pure GS, sample #D2 (Supplementary Fig. 16b), which shows that assembly by the EDPSE method also can produce well-ordered mesocrystals in confined space. The higher crystalline quality of mesocrystals of sample #D1 compared to sample #D3 confirms that a sufficient addition of OA is essential to assemble large and well-ordered mesocrystals.

The time-resolved scattering data in Fig. 5f–h show that the mesocrystals start to form in the GS/PS dispersions (samples #D1 and #D3) at a critical concentration of $c_{DL} = 17\ mg\ mL^{-1}$, but (meso)crystallization in the pure GS dispersion (sample #D2) is initiated at a $c_{DL}$ of $230\ mg\ mL^{-1}$. Hence, the levitating drop studies corroborate that the poor-solvent enrichment is essential to induce assembly of the oleate-capped nanoparticles assemble at a relatively low particle concentration. The higher critical concentration in the levitating drop compared to the mesocrystal growth studies in the vessels can be related to the much higher solvent evaporation rate from a drop compared to a vessel, and the lower polarity of 1-pentanol compared to 2-propanol.

Figure 5f–h shows that the partial scattering invariant $Q^*$, which relates to the total scattering within the irradiated sample volume, increased with increasing time and thus particle concentration in all the three investigated systems, i.e., samples #D1–D3. The decrease of $Q^*$ shortly after the onset of mesocrystal formation is probably related to partial removal of the growing mesocrystals from the beam by accumulation at the liquid-air interface of the shrinking drop. The time between the onset of (meso)crystallization and when $Q^*$ reached its maximum (red marked area in Fig. 5f–h) is very short (6 s or less), which suggests that mesocrystal growth is very rapid in the levitating drops.

The peak area $A_{(111)}$, which relates to the amount of crystalline phase within the droplet, increases slowly during the early growth stage (red marked areas) followed by a rapid increase during the subsequent major growth stage, which corresponds to the following 45–50 s (blue marked areas in Fig. 5f–h). The asymptotic decrease of the interplanar distance $d_{(111)}$ until $A_{(111)}$ and $Q^*$ reach constant values towards the end of the major growth stage, shows that the structural changes are insignificant as crystal growth ceases (Supplementary Fig. 18). The interplanar distances $d_{(111)}$ in sample #D1 and #D3 are smaller compared to #D2, which can be attributed to the contraction of the long alkyl chains of the capping agent in the presence of a polar solvent (Supplementary Fig. 18). The interplanar distances $d_{(111)}$ increased after 300 and 500 s for sample #D1 and #D3, respectively, which can be attributed to stretching of the alkyl chains due to the removal of the polar solvent at the later stage of droplet shrinkage.

The SAXS measurements on the droplet with no added OA (sample #D3) displayed strong fluctuations of both $A_{(111)}$ and $Q^*$ after $t = {\sim}270\ s$ (Fig. 5h and Supplementary Fig. 18c). We attribute this behavior to phase separation (see insets) during a stage when the mesocrystal growth is nearly complete. Indeed, the surface of the mesocrystals retrieved from the dry beads were disordered (Fig. 5e), although the SAXS data indicates that the mesocrystals exhibit long-range order (Supplementary Fig. 16c). Hence, the SEM and SAXS data suggests that the mesocrystals are mainly disordered close to the surface but the cores of the mesocrystals are well-ordered. The mesocrystals grown from a simple GS dispersion (sample #D2) display a longer nucleation and early growth stage compared to droplets containing a PS/GS mixture (samples #D1 and #D3) (red marked areas in Fig. 5f–h), which suggests that larger amounts of nuclei/small mesocrystals were formed during this stage. The nuclei and small mesocrystals may attach and intergrow into each other during the subsequent growth stage, which can explain the abundancy of strongly twinned assemblies (Fig. 2f and Fig. 5d).

**Phase diagram and assembly pathways of EDPSE.** We have attempted to systematize the assembly information into a schematic phase diagram (Fig. 6a) and assembly pathways (Fig. 6b) for mesocrystal formation by EDPSE. The phase boundary in the simple phase diagram in Fig. 6a is related to the critical polarity of systems with $N_{PE} = 1$ and constant $V_{OA}/m_{NC}$ (samples #1–7; Supplementary Table 2), estimated by:

$$c_C(P_C) = 99.3 - 94.8P_C + 21.8P_C^2 \qquad (4)$$

The time-dependent critical concentration $c_C(t)$ for different initial $V_{GS}/V_{PS}$ values in the assembly pathways shown in Fig. 6b were estimated by combining Eq. 4 with Eqs. 1 and 2. The solid lines in Fig. 6b show the time-dependent dispersed (free) NC concentration $c_D(t)$ of samples #1–4. The assembly pathways can be predicted by extracting the intersection of the $c_D(t)$ and $c_C(t)$ curves, which coincides with the onset of the major growth stage, as well as the slope of the $c_D(t)$ curve at the intersection, which relates to the crystal growth rate ($\sim\frac{c_C}{\Delta t}$) (Fig. 6b). We show in Fig. 6b that the evaporation of hexane is associated with a decrease of $c_C$

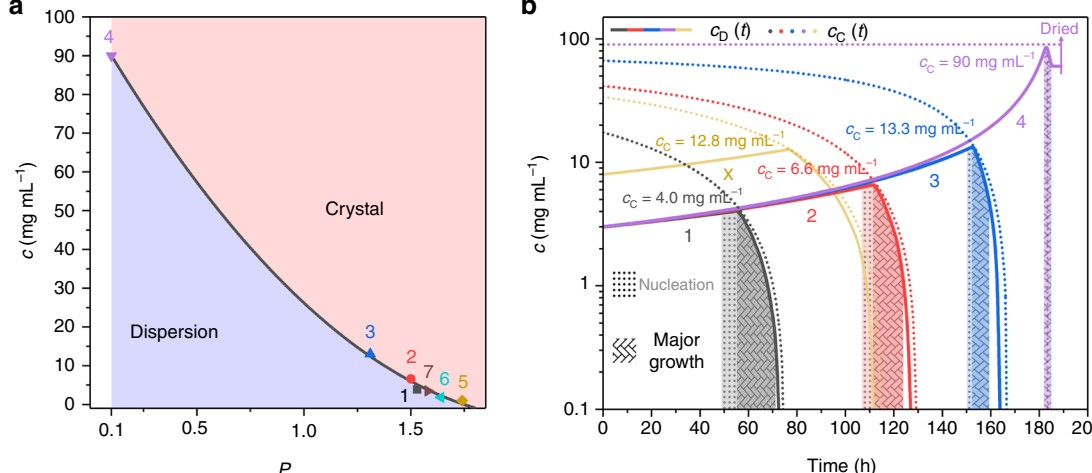

**Fig. 6** Mesocrystal phase diagram and nanocrystal assembly pathways by evaporation-driven poor-solvent enrichment. **a** Schematic phase diagram for mesocrystal formation by evaporation-driven poor-solvent enrichment. The phase boundary was determined by polynomial fitting of $c_C$ and $P_C$ for samples #1–7. **b** Nanocrystal assembly pathways for $V_{GS}/V_{PS} = 2.5$ (black), 5 (red), 12.5 (blue), ∞ (purple) and 4 (brown). The $c_D(t)$ curves of samples #1 (black), #2 (red), #3 (blue), #4 (purple), and #X (brown) are displayed as solid lines. The nucleation/early growth stage and major growth stage were marked by the dotted shaded area and braid pattern, respectively

due to the increasing $P$, while $c_D(t)$ increased due to the decreasing total volume of the dispersion ($c_D(t) = c(0) \times \frac{V(0)}{V(t)}$). Assembly is initiated when $c_D(t)$ approaches $c_C(t)$, as marked by the dotted, shaded area in Fig. 6b. During the major growth stage, $c_D(t)$ is assumed to follow the $c_C(t)$ curve. We find that the $c_C$ values of samples #1–4 obtained from the intersection of the $c_C(t)$ and $c_D(t)$ curves in Fig. 6b, correspond well with the experimentally observed values marked in Fig. 2a.

We have evaluated the predictive ability of our model and compared the experimental and calculated values for the onset and duration of mesocrystal formation for sample #X with the starting parameters $V_{GS}/V_{PS} = 4$, $c(0) = 8\ \mathrm{mg\ mL^{-1}}$, $N_{PE} = 1$ and $V_{OA}/m_{NC} = 1.6\ \mathrm{\mu L\ mg^{-1}}$ (Table 1). The $c_C(t)$ (dotted purple line in Fig. 6b) and the $c_D(t)$ curve (full purple line in Fig. 6b) for this specific composition was estimated using Eq. 4 in combination with Eqs. 1 and 2. The intersection of the two curves occurs at $c_C = 12.8\ \mathrm{mg\ mL^{-1}}$, which is quite close to the experimentally determined value of $c_C = 13.3\ \mathrm{mg\ mL^{-1}}$ (Supplementary Table 2). The assembly rate was found to be slow ($\Delta t = 24\ \mathrm{h}$), which correspond well to the gentle slope of the $c_D(t)$ curve. Hence, the EDPSE assembly pathway diagram and the associated equations are able to successfully predict the onset and duration of mesocrsytal formation and can thus be used as a tool for the assembly of large and well-ordered nanoparticle superlattices.

## Discussion

We have developed and investigated a facile approach based on evaporation-driven PS enrichment for the growth of mesocrystals with tunable size and/or morphology distribution and degree of order. The mesocrystal apparent size can be tuned from 10 μm to several hundred microns and the morphology can be controlled by means of variation of the growth parameters. The crystalline quality of the resulting mesocrystals can be controlled by both the crystal growth rate and the amount of excess OA. The EDPSE assembly process was investigated in detail by relating the solvent composition, initial nanocrystal concentration, poor-solvent enrichment rate, and excess surfactant to the onset and duration of mesocrystal growth and the size and degree of order of the assembled mesocrystals. Real-time SAXS, probing mesocrystal formation in a levitating droplet, confirmed the dependence of the crystalline quality on the presence of a destabilizing PS and excess surfactant. The longer nucleation stage of the pure GS system compared to the GS/PS system causes a large number of nuclei to intergrow and form strongly twinned mesocrystals which results in a lowered crystalline quality. The mesocrystal growth studies in both bulk and confined space reveal that EDPSE is a robust and predictable assembly method. The assembly process was facilitated by excess surfactant and could be described and predicted by a simple empirical model that is based on the evaporation-driven increase of the solvent polarity and particle concentration. The high control and predictability of the size and crystallographic quality of the EDPSE assembled mesocrystals is an important step towards large-scale production of nanocrystal superlattices from dispersions of nanocrystals coated with long-chain soft ligands such as OA, oleylamine, and trioctylphosphine. Large superlattices with iso-orientated nanocrystals have a wide range of applications in electronics, catalysis, and energy storage[12,52,53], and can also enhance the performance of current nanocrystal-based devices such as high density information storage[17,54–57] and light emitting devices[4,58].

## Methods

**Synthesis of ferric oleate**. Ferric oleate, which is used as a precursor for the NC synthesis, was synthesized from iron(III) chloride hexahydrate (98%, Sigma-Aldrich) and sodium oleate (97%, TCI) in a solvent mixture of hexane and water[59].

In a typical synthesis, 40 mmol of iron chloride and 120 mmol of sodium oleate was dissolved in a mixture solvent of 80 ml ethanol, 60 ml distilled water and 140 ml hexane, and was stirring at 70 °C for four hours. After cooling to room temperature, the upper organic layer was separated and washed three times with 30 ml distilled water. Then hexane was rotated evaporated in order to obtain ferric oleate as a dark red wax.

**Synthesis and purification of the iron oxide NCs**. The 10.8 ± 0.6 nm truncated iron oxide NCs were synthesized by high temperature decomposition of ferric oleate at 315 °C in a 1-octadecene (90%, Sigma-Aldrich)/1-hexadecene (90%, TCI) mixture, as described previously[39]. Under a moderate inert gas flow, 5 mmol ferric oleate, 0.7 mmol OA (99%, TCI), and 0.7 mmol sodium oleate were dissolved in 25 mL 1-octadecene and 5 mL 1-hexadecene in a 100 mL three-neck flask. The thermal decomposition of the precursor starts with degassing the solution at 140 °C for 30 min. This step also helped to remove volatile impurities from the reaction mixture, which may cause violent splashing and temperature fluctuation in subsequent stages. Under a blanket of inert gas, the mixture was heated to 315 °C at 3 °C min⁻¹ and kept at this temperature for 30 min. After rapidly cooling the reaction to room temperature, the mixture was transferred to a 250 mL cone flask and shaken for 30 min with 150 mL ethanol (99.5%, SOLVECO). The ethanol phase was decanted subsequently and the remaining non-polar phase, containing the NCs was diluted by adding 3–5 mL toluene (100%, VWR) and shaken vigorously for 30 min. After four washing cycles with ethanol, the dispersion was vacuum dried at 60 °C and stored at 4 °C under inert gas for further use. Approximately 900–1000 mg tar-like product was obtained, containing ca. 40–50% of iron oxide. For the preparation of high-quality mesocrytals, the as synthesized NCs should be used within 3 months, otherwise the quality of the resulting mesocrystals may start to deteriorate. Five ethanol purification cycles of the reaction mixture yielded approximately 900–1000 mg of a tar-like product, which contained ca. 43% of iron oxide.

The tar-like synthesis product was further purified by repeated re-dispersion and magnetic separation in mixtures of a non-polar solvent and 1-pentanol (99%, Sigma-Aldrich). The ethanol-washed tar-like synthesis product (100 mg) was finally dispersed in 1 mL of a non-polar solvent, shaken and then sonicated for 15 min, respectively. We then added 15 mL 1-pentanol to the dispersion and the mixture was first shaken and then sonicated for 15 min, respectively. The iron oxide NCs were separated using a magnet and the liquid was decanted. The washing process was repeated four times resulting in a purified black powder which contained 91 wt% of iron oxide and 9 wt% OA corresponding to an OA coverage of 2.0 molecules nm⁻². All chemicals were used without further purification and purchased from commercial sources.

**Mesocrystals growth**. Dispersions of oleate-capped iron oxide NCs with an initial NC concentration $c(0) = 3.0\ \mathrm{mg\ mL^{-1}}$ were prepared by mixing 45 mg of the purified NC powder with a 15.0 mL solvent mixture of a GS (hexane, 99%, Merck) and PS (2-isopropanol, 100%, VWR) with volume ratios of $V_{GS}/V_{PS} = 2.5, 5, 12.5$, and ∞ (∞ means pure GS). Then, 0 to 144 μL of OA (with a density of 0.9 mg μL⁻¹) were added to the dispersions at certain volume to mass ratios, keeping the ratio of the volume of added OA to NC powder weight $V_{OA}/m_{NC}$ at desirable values. Detailed growth parameters were listed in Table 1.

The NC dispersions in the OA-containing solvent mixtures were then shaken and sonicated for 30 min, respectively, followed by the removal of aggregates by filtration using a 0.2 μm glass fiber syringe filter. The sonicated and filtrated dispersions were then transferred into a cylindrical 20 mL glass vessel with a 2 cm² opening area. The solvent evaporation rate was controlled by maintaining a constant temperature (25 °C) and covering the top of the glass vessel with one, two or four layers of PE foil ($N_{PE} = 1, 2$ or 4, 13 μm thickness, Trajmi AB). Mesocrystals formed after 75−300 h of evaporation, depending on the evaporation rate and the initial solvent composition, and continued to grow until the liquid phase was depleted from nanoparticles. The mesocrystals can be easily collected by gentle shaking or by applying a weak magnetic field. The measurement of the evaporation rate of pure solvents (hexane and 2-propanol) was performed under the same condition.

**Characterization**. The height of the initial liquid level was measured by a Diesella digital caliper (Diesella, Sweden). The evaporation rate was determined by weight loss using an electronic balance (readability 0.1 mg, ACCULAB ATL-224, Germany).

Fourier transform infrared spectroscopy (FTIR) was performed on a Varian 670-IR spectrometer (Agilent, USA) and thermogravimetry (TG) analysis was performed on a Discovery TGA 1 (TA Instrument, USA).

SEM and HRSEM were performed on a JEOL JSM-7000F microscope (JEOL, Japan), equipped with a Schottky-type FEG and operated at 15 kV. Mesocrystals for SEM analysis were collected from the bottom of the vessel and washed carefully with 2-propanol, and then subjected to UV ozone treatment (BioForce Nanoscience, USA) to remove all organic residue prior to imaging. Some mesocrystals were attached onto a silica substrate and cross section polished by a perpendicular ionized argon beam at 5 kV accelerating voltage for 15 h on a JEOL CP-09010 cross section polisher (JEOL, Japan). TEM and high resolution TEM (HRTEM) of individual NCs were performed on a JEOL JEM-2100F microscope

(JEOL, Japan) operated at 200 kV, by drop-casting the purified NC dispersion in toluene onto a copper grid. Fast Fourier Transform (FFT) of the HRSEM/HRTEM was performed by ImageJ (https://imagej.nih.gov/ij/). The apparent size was the multi-measurement average of individual mesocrystal sizes measured by Nano Measurer (Department of Chemistry, Fudan University) from SEM images, and the individual mesocrystal was always measured along the longest direction to minimize the measuring error.

Optical microscopy was performed on a Nikon Eclipse FN1 microscope with a 10× eyepiece and 10×/50× long working distance objective (Nikon, Japan) equipped with a 2 megapixel CCD sensor for image recording (Kappa Zelos-02150C GV, Kappa Optronics GmbH, Germany).

SC-XRD characterization was carried out on a Bruker D8 VENTURE single-crystal X-ray diffractometer (Bruker, German) using Mo Kα radiation ($\lambda_{K\alpha 1} = 0.07093$ nm) on a single mesocrystal mounted on a glass fiber. The CCD detector was rotated around the single mesocrystal at a distance of 50 mm and in steps of 15° with an acquisition time of 60 s for each image.

DLS, zeta potential, and electrophoretic mobility measurements of the NC dispersions were performed on a Zetasizer Nano ZS (Malvern, UK). DLS measurements were carried out on NCs dispersed in hexane with a concentration of 0.02 mg mL$^{-1}$. For zeta potential and electrophoretic mobility measurements, we used NCs dispersions with a concentration of 0.1 mg mL$^{-1}$ in chloroform.

**Real-time SAXS measurement.** Samples #D1–D3 were obtained by dispersing purified NCs powder in a mixture of octane (98%, Sigma-Aldrich), 1-pentanol, and OA. Detailed growth parameters were listed in Table 2. Scattering experiments of #D1–D3 were carried out at the P03 beamline at DESY, Hamburg, Germany. The data was recorded by a Pilatus 1 M detector covering a range of 0.11 nm$^{-1}$ < $q$ < 3.59 nm$^{-1}$. The time-resolved data was acquired with an exposure time of 0.5 s per frame resulting in a time resolution of 0.9 s. The 2D data was reduced and integrated to a 1D pattern using the program DPDAK[60]. The colloidal droplet was injected into an acoustic levitator (model 13K11, tec5, Oberursel, Germany) and irradiated by a square shaped beam with a spot size of 20 × 20 μm$^2$ and a wavelength $\lambda = 0.96$ Å. The partial scattering invariant $Q^*$ was obtained by integrating $\int_{q_{min}}^{q_{max}} I(q)q^2 dq$ in the range 0.11 nm$^{-1}$ < $q$ < 3.59 nm$^{-1}$. We fitted a pseudo-Voigt function to the (111) peak of our crystalline assemblies to get information on the value of $d_{(111)} = \frac{2\pi}{q_c}$ with $q_c$ being the peak center and the peak area $A_{(111)}$, which gives insight to the rate of crystal growth. The structure factor $S(q)$ in Fig. 1c and Supplementary Fig. 16 has been obtained by $S(q) = \frac{I(q)}{P(q)}$ with $I(q)$ being the measured intensity and $P(q)$ being the form factor of the NCs. The form factor was taken from the first frame of the time-resolved measurements when the dispersions were free of aggregates and the contribution to the scattered intensity can be assumed to be purely by the form factor.

The droplet was simultaneously observed with a microscope camera to correlate the onset of crystallization to the concentration. The video was decomposed into image frames with the program VirtualDub (http://www.virtualdub.org/). The radii $a$ and $c$ of the oblate ellipsoidal droplet were obtained by analyzing the image frames with ImageJ and the droplet volume was calculated by $V_{DL}(t) = \frac{4}{3}\pi a^2 c$. The time-dependent concentration can be calculated by $c_{DL}(t) = \frac{c_{DL}(0)V_{DL}(0)}{V_{DL}(t)}$ with $c_{DL}(0)$ and $V_{DL}(0)$ as the initial concentration and volume, respectively. For #D1 and #D2, due to the existence of OA, the droplet size remained unchanged after the evaporation of octane and 1-pentanol after ~600 s and ~400 s, respectively (Supplementary Fig. 18). For #D3, due to the lack of OA, the droplet kept shrinking for more than 20 min and eventually bursted in the ultrasonic wave at a sufficiently small size. To compare mesocrystal growth in a levitating drop to growth in a glass vessel where further solvent evaporation was prevented after the major growth stage, we analyzed the first 700 s (Supplementary Fig. 18) and discuss mainly the major growth stage during the first 300 s for #D1 and #D3, and 400 s for #D2 (Fig. 5f–h).

## Data availability

The raw datasets generated during and/or analysed during the current study (SEM, SAXS, OM, photos, SC-XRD, TEM, IR, TG, DLS, Zeta, etc.) are available at Figshare (https://figshare.com/) with a DOI of https://doi.org/10.17045/sthlmuni.9488804. Source data underlying Figs. 1c, e, 2a, f, 3a, c, 4a–b, 5f–h, 6a–b, and Supplementary Figs. 1b, 2, 3, 5, 6, 9, 12, 14, 16 and 18 are provided as a Source Data file.

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

## Acknowledgements

We acknowledge the Swedish Research Council (VR) for funding this work. We thank Kjell Jansson, Yuan Zhong, Xia Wang and Bin Wang and for assistance with SEM/TEM measurements, and Pierre Munier, Michael Agthe, Mo Segad, and Yingxin Liu, for assistance with the acquisition of SAXS data, and Mo Segad and Sun Yu for assistance with SAXS data analysis, and Andrew Kentaro Inge for the assistance with the SC-XRD measurement, and Per Jansson for technical assistance with the acoustic levitator. We are also grateful to Tomás Plivelic from the CoSAXS beamline at MAX IV for providing the ultrasonic levitator, and Andrija Lazic for providing materials for 3D artwork. We acknowledge DESY (Hamburg, Germany), a member of the Helmholtz Association HGF, for the provision of experimental facilities. Parts of this research were carried out at PETRA III and we would like to thank Wiebke Ohm for assistance in using the P03 beamline. The research leading to this result has been supported by the project CALIPSOplus under the Grant Agreement 730872 from the EU Framework Programme for Research and Innovation HOR-IZON 2020. Open access funding provided by Stockholm University.

## Author contributions

Z.-P.L. conceived and designed the study, and analysed the materials. Z.-P.L., M.K. and L.B. designed the time-resolved SAXS levitating drop study. M.K. and Z.-P.L. performed SAXS measurement and analysed the SAXS data. Z.-P.L. M.K. and L.B. interpreted the data, wrote the paper and revised the manuscript.

## Additional information

**Competing interests:** The authors declare no competing interests.

