## [Peer Review File · Nature Communications]

Reviewers' Comments:

Reviewer #1:

Remarks to the Author:

This experimental paper presents self-assembly experiments of colloidal nanocubes into large supra-crystals through a method coined "evaporative poor solvent enrichment". Two methods are classically used to assemble nanocrystals: evaporation of a colloidal dispersion or destabilization via the addition of a poor solvent which impact the interaction potential between nanocrystals and provokes assembly. This work combines both methods: a colloidal dispersion of nanocrystal in a good solvent is mixed with a poor solvent with a lower vapor pressure. The good solvent evaporates over a few days and ordered super-lattices are formed. The authors first describe their method and study the effect of several experimental parameters such as the volume ratio of good and poor solvent, the initial concentration in nanocrystals, the effect of addition of oleic acid or the speed of evaporation. The morphology of the super-lattices are observed with electron microscopy (SEM and TEM). These experiments are complemented with in situ SAXS experiments on levitating evaporating droplets which unravel the structure of the dispersion during the evaporation.

I am pretty enthusiastic about the paper which i find worth publishing in Nature Communication. It has several very interesting aspects:

- it focuses on the physical chemistry of the drying process and thoroughly explores this aspect with a reference system. This is not common in the literature. Most papers often focus on the formation of new exotic self-assembled structures without trying to rationalize the assembly process. In this sense, the paper is important for its focus on a usually neglected issue.
- there is a lot of work in the paper where several effects have been explored extensively.
- the analysis of the results is particularly deep with a modelling effort of the evaporating process and an attempt to gain predictive power through a phase diagram approach. This typical physical chemistry endeavour is all too rare in the nanotechnology community and i find this aspect particularly novel and insightful.
- all the experimental methods, from the synthesis and purification of the nanocrystals are extensively described and very detailed. I have no doubt that someone trying to reproduce the results would be able to do so.

I thus recommend publication but i have a few comments which could improve the quality of the paper if the authors could address them.

- characterization of the individual nanocubes should be present in the main manuscript and not in SI. At least one TEM image should be included in figure 1 for example.
- the effect of Npe is not clearly detailed in the paper. It would interesting to describe in more details the differences in the supra-crystals obtained with the variation of this parameter. Are the shape and structure of impacted by Npe ?
- the effect of [OA] on the self-assembly is rationalized in terms of electrostatic interaction between nanocrystals. I have several questions about this. First, the zeta potential is measured in chloroform while the assembly is not performed in this solvent. Why the measurement has not been made in the same solvent ? Could the result be different in alkane ? Another concern is about ruling out depletion interactions on the argument that the concentration in OA is not high enough. I agree at the beginning of the reaction but since the solvent evaporates, the concentration increases and this effect can start to be relevant. Could the authors comment on this ? On figure 4b the concentration in OA should be displayed in the caption instead of the sample number which is not informative.
- in the method section, a filtration step using a 0.2 glass fiber is described. I am concerned that this step might modify the concentration in nanocrystals since large aggregates which are likely to form when the good and bad solvent are mixed will be stuck in the fiber. Hence, the real concentration in

NC might be different than the initial one and the difference between the two might depend on the solvent/antisolvent ratio and property. Why is the dispersion filtered ? Have the authors considered the possible biases that this step might introduce in their experiments ?

Benjamin Abécassis

Reviewer #2:

Remarks to the Author:

The manuscript of Bergström et al report the self-assembly approach based on the evaporation of good and bad solvent mixtures. In some sense, this approach bridges two known approaches - evaporation, that allows the formation of periodic nanoparticles films and controllable oversaturation, that results in the formation of faceted periodic structures. I really like the idea of this study. It is definitely fits well into portfolio of manuscripts published in Nature Communication. There is one particular cool experiment (SAXS on levitation droplet) that is very interesting (it allows to minimize the contribution of the substrate) and is worth of publishing by itself. However, my recommendation will to reconsider after major revision. It is really hard to follow this manuscript. Also important information that can be obtained in the conducted experiments is missing. The detailed comments are provided below.

1. What is the focus of this manuscript - introduction of solvent-nonsolvent approach or effect of ligand? First part of manuscript is on hexane/1-propanol, second on effect of ligands and third is on octane-pentanol levitating droplets... All experimental are justified, but the experiments are disconnected.
2. Introduction - why evaporation rate is better controlled by PE membraned placed on the top of the vial than by pressure? What is the scientific explanation? Can you, please, add experimental data demonstrating that?
3. Why hexane-1-propanol mixture has been selected?
4. Mean size of the crystals is not meaningful. There is a huge size distribution and mesocrystals have different shapes and sizes along different directions. In my opinion, the size analysis can be completely removed from the text.
5. The authors state that their approach allows to achieve better control as compared to controlled oversaturation. I disagree with this point. The provided data do not indicate that. However, I agree that the proposed approach allows expediting of the self-assembly process to the very large extend.
6. All figure caption are hard to follow.
7. The effect of different experimental parameters (solvent ratio, etc) on the interparticle spacings should be added and compared with those for periodic films and mesocrystals obtained by controllable oversaturation.

Reviewer #3:

Remarks to the Author:

The authors present a facile approach using the solvent evaporation-driven method to grow mesocrystals containing truncated iron oxide nanocube based building blocks with tunable size and/or morphologies. The growth process was systematic studies and various conditions were carefully investigated. This work could be a paradigm, demonstrating super crystal growth.

My only question is about the superstructure characterization. The fcc superstructure concluded from the SAXS data were determined as tilted face-to-face packing structure (i.e. the cubic building blocks

are tilted by 45° in x, y, and z directions). I am not sure whether such a configuration described as "face-to-face along [111] direction" (stated on page S20) in this case is true or not. Isn't "face-to-face along with a [110]-supercrystal direction"? Do the authors mean that the "[111] direction" is "the building blocks' [111] direction"? In addition, this part (superstructure of the mesocrystals) contains a significant insight from the study of these assembled mesocrystals including samples D1, D2, and D3. I strongly suggest that the authors should move this discussion from the supporting part to the manuscript.

In conclusion, I recommend a publication of this manuscript once the questioned point is resolved.

Reviewer #1:

Q1: *Characterization of the individual nanocubes should be present in the main manuscript and not in SI. At least one TEM image should be included in figure 1 for example.*

A1: We have redesigned the layout of Fig. 1 and made the HR-TEM image of an individual nanocube larger with better resolution and clearer lattice fringes.

Q2: *The effect of N_{PE} is not clearly detailed in the paper. It would be interesting to describe in more detail the differences in the supra-crystals obtained with the variation of this parameter. Are the shape and structure impacted by N_{PE} ?*

A2: We observed a slightly increased mean supra-crystal size and crystalline quality due to the slower growth rate at higher N_{PE} . This result is reasonable, since c_C and P_C , which have the most significant impact on the crystallization process, was unaffected by N_{PE} . We did not explore extremely slow evaporation rates, using $N_{PE} > 4$, due to the very long duration (several months) for one experiment, and because it is reasonable to assume that the outcome would be similar to $N_{PE} = 4$. We have expanded/revised the discussion of the effect of N_{PE} : line 182 to line 190, “*At increased from 11 h to 22 h with*

increasing N_{PE} while c_C and P_C were unaffected... .., with an apparent size of about 100 μm with well-defined facets without boundaries or distortions on the (110) facet that was grown at a very low growth rate (sample #9, $N_{PE}=4$) The mesocrystals grown using $N_{PE} = 1, 2, \text{ and } 4$ all displayed a $c2mm$ symmetry, as shown by the FFT patterns at different magnifications in Fig. 3d.”.

Q3: *The effect of [OA] on the self-assembly is rationalized in terms of electrostatic interaction between nanocrystals. I have several questions about this. First, the zeta potential is measured in chloroform while the assembly is not performed in this solvent. Why the measurement has not been made in the same solvent? Could the result be different in alkane? Another concern is about ruling out depletion interactions on the argument that the concentration in OA is not high enough. I agree at the beginning of the reaction but since the solvent evaporates, the concentration increases and this effect can start to be relevant. Could the authors comment on this? On figure 4b the concentration in OA should be displayed in the caption instead of the sample number which is not informative.*

A3: We appreciate the insightful comments regarding how oleic acid, OA, influences the assembly process. As stated in the manuscript, there are several reports on the effect of OA on the assembly of well-ordered superlattices but there is no consensus or strong proof regarding how OA influence the interactions between the nanoparticles.

About the first question, we indeed tried to perform the zeta measurement in pure hexane or hexane/2-propanol but the measurements were unreliable. Using chloroform as the solvent with a bias voltage of 20 V resulted in reproducible and reliable measurements. We refer to other works (Shevchenko, Elena V., et al. "Structural diversity in binary nanoparticle superlattices." Nature 439 (2006): 55.) where different solvent systems were used for superlattice growth and zeta measurements. We have revised the method part from line 458 to 460 as “*For zeta potential and electrophoretic mobility measurements, we used NCs dispersions with a concentration of 0.1 mg mL⁻¹ in chloroform.*”

For the second question, we agree that depletion interactions cannot be ruled out in some samples. For example, in sample 4 where only hexane was used as a solvent, the OA volume fraction (x_{OA}) was estimated to be 15.1% at the onset of mesocrystallization. In “Chen C J, Chiang R K, Jeng Y R. Crystallization and magnetic properties of 3D micrometer-scale simple-cubic maghemite superlattices. The Journal of Physical Chemistry C, 2011, 115(37): 18142-18148”, they showed that the critical volume fraction of OA for depletion driven assembly of iron oxide nanocube dispersions in hexane was $x_{OA} = \sim 15\%$. Based on their work, we assumed that depletion interactions in all other systems of our work, where x_{OA} is between 0.6% and 3%, has a minor impact on the self-assembly/mesocrystallization. The discussion in the revised manuscript has been modified manuscript from line 244 to 248, “..... (x_{OA} always less than 2.1%, see Supplementary Table 2). However, the critical OA volume fraction x_{OA} in a dispersion

that only contain GS (sample #4) was as high as 15.1% when the crystallization took place (see Supplementary Table 2), which suggest that the assembly in this (non-EDPSE) system could be driven by attractive depletion forces caused by the high OA concentration rather than by the increasing volume fraction of the NCs.”

We added the parameter x_{OA} in Table S2 and have furthermore revised Fig. 4b and added the initial OA concentrations.

Q4: In the method section, a filtration step using a 0.2 glass fiber is described. I am concerned that this step might modify the concentration in nanocrystals since large aggregates which are likely to form when the good and bad solvent are mixed will be stuck in the fiber. Hence, the real concentration in NC might be different than the initial one and the difference between the two might depend on the solvent/antisolvent ratio and property. Why is the dispersion filtered? Have the authors considered the possible biases that this step might introduce in their experiments?

A4: We are grateful to this insightful question. The major reason we filtered the dispersion was to ensure that no dust or other kind of impurities coming from the environment could affect the mesocrystallization. We carefully checked for any black residue on the filter surface to ensure that the (black) oleate-capped iron oxide nanocrystals passed through the filter. The phase diagram in Fig. 6a (see the figure below which is a zoom in of Fig. 6a) also suggests that all the dispersions with an initial concentration $c(0)$ of 3 mg/mL and $V_{GS}/V_{PS} \geq 2.5$ is at a state where no aggregation/mesocrystallisation is expected.

Reviewer #2:

Q5: What is the focus of this manuscript - introduction of solvent-nonsolvent approach or effect of ligand? First part of manuscript is on hexane/1-propanol, second on effect of ligands and third is on octane-pentanol levitating droplets... All experimental are justified, but the experiments are disconnected.

A5: We are aware that the manuscript contains much information and may be difficult to digest. The main focus of the manuscript is to introduce the novel evaporation-driven poor solvent enrichment (EDPSE) method for controlled growth of well ordered mesocrystals. However, several parameters, including the addition of oleic acid (OA) has to be controlled and optimized to ensure that the size and degree of order of the mesocrystals obtained using the EDPSE method is maximized. The levitating droplet experiments were performed in a different poor solvent/good solvent pair to increase the evaporation time and thus enable time-resolved studies to be made. We have done our best to clarify the different parts of the manuscript with subheadings.

Q6: *Introduction - why evaporation rate is better controlled by PE membrane placed on the top of the vial than by pressure? What is the scientific explanation? Can you, please, add experimental data demonstrating that?*

A6: We have modified and expanded the description of how the solvent is removed from a vial covered with a PE membrane. In short, the solvent removal is driven by evaporation, but the rate is controlled by the diffusion through the membrane. The mass transfer of the solvent vapor becomes a slow diffusion process through the membrane (quasi equilibrium and isothermal) rather than a fast evaporation process due to reduced pressure (nonequilibrium).

Hence, at a constant temperature and pressure, the diffusion constant is only determined by the thickness of the membrane x ($P \propto 1/x$). In an isothermal condition with ventilation, $p_a^0 \approx 0$; we obtain:

$$q = PA_0 p_a$$

where p_a is the saturated vapor pressure of component a , which is a function of only temperature. This means we can set the mass transfer rate q freely by modifying the membrane thickness x and the vessel opening area A_0 . The detailed physical model can be found in Supplementary method 2. Figure S3b–d shows that the mass transfer rate of both hexane and 2-propanol were basically constant in an isothermal environment. By doubling the number of PE layers, the mass transfer rate decreased by 1.5 times, while the ratio of the evaporation rates of the two solvents remain unchanged (Figure S3b–d), which confirms the diffusion model of solvent vapor removal.

In contrast, rapid evaporation under reduced pressure will lead to a fast temperature decrease, where the mass transfer process is difficult to describe. As shown in Figure S3a, in an open evaporation system, the evaporation rate of hexane and 2-propanol decreases rapidly due to evaporative cooling.

We have revised the description and discussions of the solvent removal process in the revised manuscript: **line 66 to 67** “.....and the diffusion rate through one or several layers of a polyethylene (PE) membrane that covered the opening of the vessels”, and from **line 92 to line 101**, “.....with the evaporation rate decreasing with time due to the cooling caused by the rapid, unconstrained evaporation.....because hexane

diffuses much faster than 2-propanol through the hydrophobic PE membrane.....The removal of solvent from the dispersions is thus driven by the evaporation but the rate is primarily controlled by the diffusion rate of hexane through the PE membrane.” The Supplementary method 2 in SI was also changed slightly.

Q7: *Why hexane-2-propanol mixture has been selected?*

A7: The solvent pair should be fully miscible, display a significant difference in polarity (where the nonpolar solvent is a good solvent and the more polar solvent is the poor solvent) and finally, the nonpolar solvent should have a significantly higher partial vapor pressure at room temperature compared to the more polar solvent. Hexane is a good solvent for dispersing OA-capped nanoparticles and it has been extensively used in colloidal self-assembly. Hexane has furthermore a relatively high partial vapor pressure (evaporation rate) and high diffusion rate through the PE membrane. Moreover, hexane has a low toxicity.

The boiling point of the poor-solvent should be higher than the boiling point of hexane (b.p. 68 °C). In addition to 2-propanol (b.p. 82.5 °C), we have also tested other miscible solvents such as ethanol (b.p. 78 °C) and 1-pentanol (b.p. 138 °C) as the poor-solvent. The evaporation rate of ethanol is only ten times slower than the evaporation rate of hexane, which makes it more difficult to predict the behavior. 1-pentanol has a sufficiently high b.p, but the interaction between 1-pentanol and hexane will decrease the vapor pressure and the evaporation rate of hexane, causing the growth time to increase. 1-pentanol can certainly be used but the resulting mesocrystals exhibit smaller and more irregular morphologies compared to the 2-propanol analogues, at identical growth parameters ($V_{GS/NS} = 5$, $c_0 = 3$ mg/mL, $V_{OA}/m_{NC} = 1.6$ μ L/mg, $N_{PE} = 1$, as shown in the figure below). The polarity of alcohols with longer alkane chains is too low to efficiently destabilize the oleate-capped nanocrystals.

For the above mentioned reasons, the mixture of hexane and 2-propanol has proven to be the most suitable system for our system so far. We have expanded the discussion related to the choice of the solvent pair in the manuscript from line 88 to line 91,

“Hexane and 2-propanol were selected from a range of GSs and PSs as the optimal GS and PS pair, mainly due to their mutual miscibility, and large difference in vapor pressure and polarity. The low toxicity and relatively low costs of these solvents were also important features of relevance for large-scale applications.”

Q8: *Mean size of the crystals is not meaningful. There is a huge size distribution and mesocrystals have different shapes and sizes along different directions. In my opinion, the size analysis can be completely removed from the text.*

A8: Indeed, the size distribution is quite wide and the measurement can be somewhat arbitrary due to arbitrary selection of the mesocrystal direction. We have always used the longest axis to minimize the measuring error. From the statistical point of view, the wide size distribution is an intrinsic property, and has nothing to do with the accuracy of the measurement. Actually, samples with smaller size (e.g. sample #1, #4, #5) have quite narrow size distributions.

Although obtaining mean sizes from SEM images can be problematic, the size is still an important property for superlattices and mesocrystal, and it is possible to determine significant size differences among different samples even from the SEM/OM images. We agree that the term “mean size” is not well defined. We use the term “apparent size” in the revised manuscript and revised the sentence from line 444 to 447 as *“The apparent size was the multi-measurement average of individual mesocrystal sizes measured by Nano Measurer (Department of Chemistry, Fudan University) from SEM images, and the individual mesocrystal was always measured along the longest direction to minimize the measuring error.”*

Q9: *The authors state that their approach allows to achieve better control as compared to controlled oversaturation. I disagree with this point. The provided data do not indicate that. However, I agree that the proposed approach allows expediting of the self-assembly process to the very large extend.*

A9: We assume that the term “controlled oversaturation” (which we did not use in our manuscript), is equivalent to, “poor solvent destabilization” in line 194, which includes liquid/gas phase diffusion with a two or three-layer set-up. We have in fact used the three-layer liquid phase diffusion method for mesocrystal growth and indeed observed large mesocrystals. However, the reproducibility of the three layer set-up is not as good as the EDPSE method. For example, one more drop of the buffer layer may postpone the crystallization time, or a slightly disturbance on the interface of different layers may lead to very different results. In the EDPSE method, the good and poor solvents were mixed in advance, therefore the effect of experimental operation was minimized. Most importantly, as far as we are aware, the conventional “poor solvent destabilization” assembly methods are unable to provide precise information about the critical concentration or crystallization rate, and is thus less controlled compared to the EDPSE method presented in this study.

Q10: *All figure caption are hard to follow.*

A10: We have revised the figure captions and figure legends.

Q11: *The effect of different experimental parameters (solvent ratio, etc) on the interparticle spacings should be added and compared with those for periodic films and mesocrystals obtained by controllable oversaturation.*

A11: We have determined the interparticle spacings from the time-resolved SAXS experiment, shown in Figure S18 ($d_{(111)}$). Although the changes of interparticle spacings during solvent evaporation is an important field of study (see e.g. Josten E, Wetterskog E, Glavic A, et al. Superlattice growth and rearrangement during evaporation-induced nanoparticle self-assembly. Scientific reports, 2017, 7(1): 2802.), a detailed study and comparison with mesocrystals grown on substrates is outside the scope of this study. We have included a short discussion of the interparticle spacing in the main text, from line 300 to 305, “The interplanar distances $d_{(111)}$ in sample #D1 and #D3 is smaller compared to #D2, which can be attributed to the contraction of the long alkyl chains of the capping agent in the presence of a polar solvent (Supplementary Fig. 18). The interplanar distances $d_{(111)}$ increased after 300 and 500 seconds for sample #D1 and #D3, respectively, which can be attributed to stretching of the alkyl chains due to the removal of the polar solvent at the later stage of droplet shrinkage.”

Reviewer #3:

Q12: *My only question is about the superstructure characterization. The fcc superstructure concluded from the SAXS data were determined as tilted face-to-face packing structure (i.e. the cubic building blocks are tilted by 45° in x, y, and z directions). I am not sure whether such a configuration described as "face-to-face along [111] direction" (stated on page S20) in this case is true or not. Isn't "face-to-face along with a [110]-supercrystal direction"? Do the authors mean that the "[111] direction" is “the building blocks' [111] direction”? In addition, this part (superstructure of the mesocrystals) contains a significant insight from the study of these assembled mesocrystals including samples D1, D2, and D3. I strongly suggest that the authors should move this discussion from the supporting part to the manuscript.*

A12: We agree that the description “face-to-face along [111]” can be misleading. Indeed, the faces of the NCs are not fully overlapping and the above mentioned description of the NC packing can be therefore misleading. However, the faces of the NCs are partly overlapping when looking along the [111] direction of the superlattice. We have therefore changed the text in the SI accordingly and added a sentence in the main text in line 270 as “....., and partly overlapping in a face-to-face arrangement along the [111]-superlattice direction”. Additionally, we have revised the figure caption of Figure S17 to point out the viewing direction of the superlattice.

We have considered to follow the advice to move more structural discussion from the SI to the main text but have decided to let this part remain in the SI to not disrupt the focus on the novel EDPSE assembly method.

Besides we have also changed some wording and fixed some format issues according to the checklist. All the changes were highlighted in the manuscript. The supplementary information was also revised slightly. The legend of supplementary movie 1 is “Images and corresponding SC-XRD patterns of the rotating single mesocrystal in Fig. 1d of sample #9 ($N_{PE} = 4$) obtained from X-ray diffractometer”.

Reviewers' Comments:

Reviewer #1:

Remarks to the Author:

I thank the authors for their modification of the manuscript which, in my opinion, is ready for publication.

Reviewer #2:

Remarks to the Author:

The authors satisfactorily addressed my comments. I do not fully agree with some of their statements; however, I do not feel I have enough experimental supports to insist. I think that the revised manuscript can be published in Nature Communication. However, while the authors attempted to quantify a number of experimental parameters, the study is a bit empirical. I suggest, that the authors conduct TGA analysis of the NPs under different reaction conditions, especially when they discuss the effect of OA on the assembly. The numbers of 2.1% and 15% - do they relate to extra ligands introduced to the NPs or they describe the total amount of OA in the samples? This is an important set of data, since the initial concentration of ligands determines the effect of extra ligands.

Dr. Lennart Bergström
Department of Material and Environmental Chemistry,
Stockholm University, SE-106 91 Stockholm
Tel: 08 16 23 68
Mobil: 070 517 99 91
E-mail: lennart.bergstrom@mmk.su.se.

Reviewer #2:

Q1: The authors satisfactory addressed my comments. I do not fully agree with some of their statements; however, I do not feel I have enough experimental supports to insist. I think that the revised manuscript can be published in Nature Communication. However, while the authors attempted to quantify a number of experimental parameters, the study is a bit empirical. I suggest, that the authors conduct TGA analysis of the NPs under different reaction conditions, especially when they discuss the effect of OA on the assembly. The numbers of 2.1% and 15% - do they relate to extra ligands introduced to the NPs or they describe the total amount of OA in the samples? This is an important set of data, since the initial concertation of ligands determines the effect of extra ligands.

A1: The comment relate to the important question of how to discriminate between the surface-bound OA and the free OA. This is indeed an important question that was not fully discussed in the manuscript. In fact, we went to a great length to ensure that any unbound OA was removed from the oleate-capped nanocubes (NCs) by a thorough purification procedure, which is described in detail in the experimental procedure. Previous (unpublished) work has shown that insufficient purification can in fact result

in significant amounts of unbound OA remaining in the “mother” dispersion or paste. The FTIR spectra in Supplementary Figure 2a, shows that only chemically bonded OA molecules remained on the NC surface after the purification process, and the excess ligands were removed. The peak at 1710 cm^{-1} , belonging to unbound carboxylic groups, was absent in the purified NC powder, while the spectra displayed asymmetric and symmetric vibration modes of COO^- at 1397 cm^{-1} and 1517 cm^{-1} , respectively, which are indicative of surface-bound OA (L. Zhang, et al, Appl. Surf. Sci. 253 (2006) 2611–2617). The TG data of the purified NC powder in Supplementary Figure 2b, shows that the weight fraction of chemically bonded OA was 9%. By assuming a monomolecular OA surface-bound layer on a $10.8\text{ nm Fe}_3\text{O}_4$ cubic NP core, the OA weight fraction corresponds to an estimated ligand coverage on the NP surface of $2.0\text{ molecules/nm}^2$, which agrees well with previously reported OA coverage on iron oxide NPs ($1.9\text{ molecules nm}^{-2}$, Z.-P. Lv et al., ACS Nano 9 (2015) 12205–12213). Other studies (M. Klokkenburg et al., Vibrational Spectroscopy 43 (2007) 243–248; E. Dubois et al., J. Chem. Phys. 111 (1999) 7147; M. Doig et al., Langmuir 30 (2014) 186–195), have shown that the coverage of oleic acid on iron oxide surface in solution may vary from $2.0\text{--}3.5\text{ molecules nm}^{-2}$ depending on the excess ligand concentration in solution. Assuming that the absorption of oleic acid on the NC surface follows the Langmuir isotherm yields the curve below, with data (filled squares) from E. Dubois et al., J. Chem. Phys. 111 (1999) 7147,

and the surface coverage (Q) described by:

$$Q = Q_{sat} \times \frac{K \times c}{1 + K \times c} = 3.5335 \times \frac{3133c}{1 + 3133c}$$

where Q_{sat} is the saturated coverage, K is the equilibrium constant, c is the excess ligand concentration. We have included the samples #14–16, #2 and #17 (open circles), on the curve, which shows that a small amount of the added OA (ca. 0.05–0.08 μL OA per mg NCs) is expected to bond to the NC surface. The resulting decrease of the free OA concentration in the dispersion is for most of the investigated dispersions very small. For $V_{OA}/m_{NC} = 1.6 \mu\text{L mg}^{-1}$ (the ratio of the volume of added OA to NCs powder weight, in most of the investigated dispersions), the estimated reduction of the free OA concentration due to adsorption is only 5% or less. In the dispersions where a relatively small amount of OA was added (samples #14–16), we estimate that the amount of free OA is reduced with 25–10%, respectively.

The manuscript has been revised and we have made some explanations and/or changes in the main text, e.g., from line 223 to 233, “TG measurements of the purified NC powder (Supplementary Figure 2b) suggests that the OA coverage on the NP surface is 2.0 molecules nm^{-2} , which agrees well with previous reports.⁴⁹ Previous work has shown that the ligand coverage of oleate-capped nanoparticles depends on the OA concentration in solution, and can increase to 3.5 molecules nm^{-2} at OA concentration in solutions of 15mmol or higher.^{50, 51} The resulting decrease of the free OA concentration in the dispersion is for most of the investigated dispersions very small. For $V_{OA}/m_{NC} = 1.6 \mu\text{L mg}^{-1}$ (the ratio of the volume of added OA to NCs powder weight in most of the investigated dispersions, see Table 1 and 2), the estimated loss of the excess ligand concentration is only 5% or less. In the dispersions where a relatively small amount of OA was added (samples #14–16), we estimate that adsorption reduces the amount of free OA with 25–10%, respectively.”. The **Supplementary Method 1** and **Supplementary Table 2** were also revised accordingly.